# Protective Effects of Marine Alkaloid Neolamellarin A Derivatives against Glutamate Induced PC12 Cell Apoptosis

**DOI:** 10.3390/md20040262

**Published:** 2022-04-12

**Authors:** Kai Zhang, Xian Guan, Xiao Zhang, Lu Liu, Ruijuan Yin, Tao Jiang

**Affiliations:** 1Key Laboratory of Marine Drugs, Ministry of Education, School of Medicine and Pharmacy, Ocean University of China, Qingdao 266003, China; zk15065683854@163.com (K.Z.); 18838966959@163.com (X.G.); 17806022265@163.com (X.Z.); 2Marine Biomedical Research Institute of Qiangdao, Qingdao 266237, China; ll3049@yeah.net; 3Laboratory for Marine Drugs and Bioproducts, Qingdao National Laboratory for Marine Science and Technology, Qingdao 266237, China

**Keywords:** synthesis, marine alkaloids, Neolamellarin A, PC12 cells, glutamate

## Abstract

Marine alkaloids obtained from sponges possess a variety of biological activities and potential medicinal value. The pyrrole-derived lamellarin-like alkaloids, especially their permethyl derivatives, show low cytotoxicity and potent MDR reversing activity. Neolamellarin A is a novel lamellarin-like alkaloid which was extracted from marine animal sponges. We reported the synthetic method of permethylated Neolamellarin A and its derivatives by a convergent strategy in 2015. In 2018, we reported the synthesis and the neuroprotective activity in PC12 cells of 3,4-bisaryl-*N*-alkylated permethylated Neolamellarin A derivatives. In this report, another series of 15 different 3,4-bisaryl-*N*-acylated permethylated Neolamellarin A derivatives were synthesized, and the outstanding protective effects of these compounds against glutamate induced PC12 cell apoptosis were presented and discussed. These Neolamellarin A derivatives which possessed low cytotoxicity and superior neuroprotective activity may have the potential to be developed into antagonists against glutamate induced nerve cell apoptosis.

## 1. Introduction

The excitatory amino acid glutamate, which is one of the most important neurotransmitters in the central nervous system, has been indicated to be involved in rapid synaptic transmission, neuroplasticity, learning, and memory [1]. However, excessive release of glutamate can cause neurotoxicity, and further lead to acute conditions including epileptic seizures and chronic neurodegenerative disorders such as Alzheimer’s disease [2]. To date, two mechanisms have been reported for glutamate-mediated neurotoxicity. One is through competitive inhibition of cystine uptake, leading to oxidative stress [3,4,5], and the other is mediated by several types of excitatory amino acid receptors, such as *N*-methyl-*D*-aspartate (NMDA)-subtype glutamate receptor, resulting in a massive influx of extracellular Ca^2+^ [6,7].

Lamellarins, a type of marine alkaloids was first isolated from the prosobranch mollusc *Lamellaria* sp. in 1985 [8]. Now, more than 70 different lamellarins and related natural pyrrole-derived alkaloids have been reported. These lamellarins which were derived from 2-amino-3-(3′,4′-dihydroxyphenyl) propionic acid (DOPA), have a variety of promising bioactivities, such as inhibition of HIV-1 integrase and MCV topoisomerase, multidrug resistance reversal (MDR) and antitumor activity [3,4,9,10]. The structural modifications and bioactivity studies of lamellarins have been reported by several groups [11,12,13]. We found that those 3,4-diarylpyrrole-derived alkaloids which showed superior multi-drug resistance (MDR) reversal even at noncytotoxic concentrations by inhibition of P-glycoprotein (P-gp)-mediated drug efflux possessed the characteristics of high lipophilicity and low cytotoxicity [5,6,7].

Neolamellarin A (Figure 1), a 3,4-bisaryl-pyrrole structural marine alkaloid was isolated from sponge *Dendrilla nigra* in 2007 [14] and was synthesized firstly in 2009 by Arafeh and Ullah [15]. Our group has studied permethylated Neolamellarin A and its derivatives for several years. In 2015, the synthesis of permethylated Neolamellarin A and its derivatives by a convergent strategy were reported [16], and then in 2018, we reported the synthesis and the neuroprotective activity on PC12 cells of 3,4-bisaryl-*N*-alkylated permethylated Neolamellarin A analogues [17]. In this paper, the synthesis and neuroprotective activity studies on PC12 cell line of 15 novel 3,4-bisaryl-*N*-acylated permethylated Neolamellarin A derivatives were discussed.

## 2. Results and Discussion

### 2.1. Chemistry

There should be two strategies to synthesize these 3,4-bisaryl-N-acylated permethylated Neolamellarin A derivatives. One strategy is directly building the 3,4-bisaryl-N-acylated pyrrole ring by one-pot process, and the other is acylation between 3,4-bisaryl-1H-pyrrole core and phenylacetyl chloride. The latter strategy was proved to be efficient in our previous study, and the synthesis of 3,4-bisaryl-1H-pyrrole is the critical step. We have reported the synthesis of the 3,4-bisaryl-1H-pyrrole through Suzuki cross-coupling reaction between 3,4-diiodinated N-trisisopropylsilyl pyrrole and aryl boronic acid ester [16]. But the long route, high cost and low yield of this method bring difficulties in building compound libraries. Hence, in this study we first synthesized 3,4-bisaryl-1-benzyl pyrrole through one-pot AgOAc-mediated method, and then debenzylation was performed to obtain the key 3,4-bisaryl-1H-pyrrole intermediates.

The one-pot AgOAc-mediated synthetic method of building 3,4-bisaryl-1-benzyl pyrrole was studied in our previous report [17]. The debenzylation of 3,4-bisaryl-1-benzyl pyrroles **4****a**–**4e** was failed in the Pd/C, H_2_ and MeOH system (with or without catalyst) or CF_3_COOH, catalytic amount of TfOH at the reflux condition. Finally, the use of t-BuOK as a base in THF/DMSO, and oxidization by O_2_ at room temperature for 1 h gave the 3,4-bisaryl-1H pyrroles **5****a**–**5e** at the yields of 85–95% [18]. The acylation of **5****a**–**5e** with fresh acid chloride **6****a**–**6c** was achieved by using n-BuLi as a base in THF at 30 °C for 10 h to produce **1a**–**1o** at the yields of 40–50% (Figure 1).

### 2.2. 3,4-Bisaryl-N-Acylated Permethylated Neolamellarin A Derivatives as Antagonists against Glutamate-Induced PC12 Cell Death

PC12 cells are derived from rat adrenal medulla pheochromocytoma, which are widely used as an in vitro model for the neuronal apoptosis and differentiation research due to the differentiated PC12 cells induced by nerve growth factor (NGF) with the typical characteristic of the neurons in morphology and function [19,20,21]. The reports have showed that high concentration of glutamate induces PC12 cell death and different types of compounds could protect PC12 cells from glutamate-induced damage, such as sesquiterpenoids, neolignan glycosides, YC-1, and xanthoceraside [22,23,24,25,26]. Here, we used the PC12 cell model of glutamate-induced damage to evaluate the neuroprotective effect of these novel Neolamellarin A derivatives **1a**–**1o**.

Firstly, the cytotoxicity of the permethylated Neolamellarin A derivatives **1a**–**1o** on PC12 cell line at concentrations of 2.5, 5, 10, and 20 μM was evaluated, and the results showed that the cell viability was above 80% even at the concentration of 20 μM, indicating that the derivatives had low cytotoxicity on PC12 cells (Figure 2). Especially the compound **1a**–**1c** exhibited superior proliferation activities. The data were as follows: **1a** (2.5 μM 110.0%, 5.0 μM 114.9%, 10.0 μM 126.7%, 20.0 μM 143.3%), **1b** (2.5 μM 112.5%, 5.0 μM 120.0%, 10.0 μM 131.6%, 20.0 μM 142.5%) and **1c** (2.5 μM 117.5%, 5.0 μM 126.4%, 10.0 μM 144.9%, 20.0 μM 162.5%). The mechanisms of the compounds **1a**–**1c** that promote the proliferative activity of PC12 cells need to be further investigated and explore whether they can be further developed. The low cytotoxicity of these compounds on PC12 cells laid a foundation for further neuroprotective effect evaluation on glutamate-induced PC12 cell death.

We constructed the glutamate-induced PC12 cell death model by the method of the previous report [27]. Firstly, PC12 cells were inoculated into 96-well plates at a density of 5000 cells/well, and cultivated for 24 h. Cells were then treated with 8 mM glutamate, and cell viability was detected after 4 h. The results showed that the survival rate of PC12 cells decreased significantly (35% ± 3%, relative to the untreated control group). Then, we used this model to evaluate the neuroprotective activity of these permethylated Neolamellarin A derivatives **1a**–**1o** and Huperzine-A (HupA, 100 μM) was used as positive control. Compounds **1a**–**1o** were added to the glutamate damaged PC12 cells at a final concentration of 2.5, 5, 10 and 20 μM, respectively, and continually cultured for another 24 h. Cell viability was measured by MTT assay. The results in Figure 3 show that all of the 15 derivatives can effectively inhibit the apoptosis of PC12 cells (induced by glutamate-induced damage) in a concentration dependent manner. With the increase of concentration, the neuroprotective activity on PC12 cells increased. In particular, when the concentration of compounds **1c**, **1f**, **1h**, **1n**, and **1o** reached 20 μM, the cell viability achieved almost 100%.

## 3. Materials and Methods

### 3.1. Chemical Synthesis

#### 3.1.1. General

Tetrahydrofuran (THF) was distillated with Na in the presence of benzophenone under argon atmosphere. Dichloromethane was distillated by molecular sieve. Dimethyl sulfoxide (DMSO) was distillated with *t*-BuOK. All other materials and solvents were obtained from commercial sources and used without further purification. Thin-layer chromatography (TLC) was performed on precoated E. Merck silica-gel 60 F254 plates. Column chromatography was performed on silica gel (200–300 mesh, Qingdao, China). Melting points were determined on a Mitamura-Riken micro-hot stage without correction. ^1^H and ^13^C NMR spectra were recorded on the Broker AVANCE NEO and Agilent DD2 500 with 500 or 600 MHz for proton (^1^H NMR) and 125 or 150 MHz for carbon (^13^C NMR) with tetramethylsilane (Me_4_Si) as internal standard, respectively. The chemical shifts (*δ*) were expressed in parts per million (ppm) downfield, and the coupling constant (*J*) values were described as hertz. High-resolution (ESI) MS spectra were recorded using a Q TOF-2 Micromass spectrometer.

The general procedures for the synthesis of important intermediates **4a**–**4e**, **5a**–**5e** and **6****a**–**6c** can be seen in Appendix A.

#### 3.1.2. General Procedure for the Synthesis of **1a**–**1o**

*n*-BuLi (1.2 mL, 3.0 mmol, 2 mol L-1 in hexane) was added dropwise to a solution of 3,4-bisaryl-1*H*-pyrrole **5a**–**5e** (2 mmol) in anhydrous THF (10 mL) at −78 °C under argon atmosphere and stirred for 0.5 h. Then **6a**–**6c** in anhydrous THF (10 mL) was added into the reaction mixture and warmed up to room temperature whereafter. The solution continued to stirred at 30 °C for 10 h and diluted with saturated aqueous NaHCO_3_ (20 mL). The mixture was extracted with EtOAc (20 × 3 mL) and washed with brine and dried with MgSO_4_, then concentrated by vacuum evaporation to give a residue of compounds **1****a**–**1o**, which was purified by flash chromatography on silica gel (petroleum ether: ethyl acetate = 1:1).

##### (3,4-Bis(2-methoxyphenyl)-1*H*-pyrrol-1-yl)(4-methoxyphenyl)methanone (**1a**)

White solid 125 mg, yield 30.3%. m.p. 145–146 °C. ^1^H NMR (600 MHz, CDCl_3_) *δ* 7.87–7.84 (d, *J* = 8.8 Hz, 2H), 7.51 (s, 2H), 7.21–7.19 (m, 2H), 7.16–7.15 (dd, *J* = 7.1, 1.6Hz, 2H), 7.01–7.00 (d, J = 9.3 Hz, 2H), 6.87–6.85 (t, *J* = 7.2 Hz, 2H), 6.82–6.81 (d, *J* = 7.7 Hz, 2H), 3.89 (s, 3H, OCH_3_), 3.45 (s, 6H, 2OCH_3_). ^13^C NMR (150 MHz, CDCl_3_) *δ* 166.86 (C), 162.92 (C), 156.62 (2C), 132.06 (2CH), 130.41 (2CH), 128.10 (2CH), 125.46 (C), 125.38 (2C), 124.43 (2C), 120.53 (2CH), 120.39 (2CH), 113.87 (2CH), 110.86 (2CH), 55.63 (CH_3_), 55.12 (2CH_3_). HRMS (ESI) *m*/*z*: calcd for M^+^ C_26_H_24_NO_4_, 414.1700; found, M^+^ 414.1701.

##### (3,4-Bis(3-methoxyphenyl)-1*H*-pyrrol-1-yl)(4-methoxyphenyl)methanone (**1b**)

White solid 134 mg, yield 32.4%. m.p. 132–134 °C. ^1^H NMR (600 MHz, CDCl_3_), *δ* 7.85–7.83 (d, *J* = 8.6 Hz, 2H), 7.42 (s, 2H), 7.21–7.18 (t, *J* = 7.7 Hz, 2H), 7.03–7.02 (d, *J* = 8.6Hz, 2H), 6.88–6.86 (d, *J* = 7.9 Hz, 2H), 6.83–6.80 (m, 4H), 3.91 (s, 3H, OCH_3_), 3.69 (s, 6H, 2OCH_3_). ^13^C NMR (150 MHz, CDCl_3_) *δ* 166.96 (C), 163.24 (C), 159.49 (2C), 135.43 (2C), 132.14 (2CH), 129.36 (2CH), 127.94 (2C), 124.89 (C), 121.22 (2CH), 120.04 (2CH), 114.06 (2CH), 113.99 (2CH), 112.80 (2CH), 55.68 (CH_3_), 55.23 (2CH_3_). HRMS (ESI) *m*/*z*: calcd for M^+^ C_26_H_24_NO_4_, 414.1700; found, M^+^ 414.1702.

##### (3,4-Bis(4-methoxyphenyl)-1*H*-pyrrol-1-yl)(4-methoxyphenyl)methanone (**1c**)

White solid 154 mg, yield 37.3%. m.p. 120–121 °C. ^1^H NMR (600 MHz, CDCl_3_) *δ* 7.84–7.83 (d, *J* = 8.8 Hz, 2H), 7.35 (s, 2H), 7.20–7.19 (d, *J* = 8.8 Hz, 4H), 7.02–7.01 (d, *J* = 8.8Hz, 2H), 6.84–6.82 (d, *J* = 8.8 Hz, 4H), 3.90(s, 3H), 3.80(s, 6H). ^13^C NMR (150 MHz, CDCl_3_) *δ* 166.93 (C), 163.10 (C), 158.69 (2C), 132.02 (2CH), 129.76 (4CH), 127.77 (C), 126.65 (2C), 125.17 (2C), 119.35 (2CH), 113.99 (2CH), 113.84 (4CH), 55.65 (CH_3_), 55.33 (2CH_3_). HRMS (ESI) *m*/*z*: calcd for M^+^ C_26_H_24_NO_4_, 414.1700; found, M^+^ 414.1704.

##### (3,4-Bis(2-methoxyphenyl)-1*H*-pyrrol-1-yl)(3,4-dimethoxyphenyl)methanone (**1d**)

White solid 162 mg, yield 36.2%. m.p. 152–154 °C. ^1^H NMR (600 MHz, CDCl_3_) *δ* (ppm) 7.53 (s, 2H), 7.51–7.49 (dd, *J* = 8.5, 2.0 Hz, 1H), 7.41 (d, *J* = 1.6 Hz, 1H), 7.21–7.19 (t, *J* = 8.5 Hz, 2H), 7.16–7.14 (dd, *J* = 7.7, 1.7 Hz, 2H), 6.96–6.94 (d, *J* = 8.3 Hz, 1H), 6.86 (t, *J* = 7.4 Hz, 2H), 6.83–6.81 (d, *J* = 7.7 Hz, 2H), 3.97 (s, 3H), 3.95 (s, 3H), 3.45 (s, 6H). ^13^C NMR (150 MHz, CDCl_3_) *δ* (ppm) 166.85 (C), 156.61 (2C), 152.59 (C), 148.98 (C), 130.41 (2CH), 128.13 (2CH), 125.54 (C), 125.45 (2C), 124.39 (2C), 123.94 (2CH), 120.57 (2CH), 120.41 (2CH), 112.68 (CH), 110.87 (CH), 110.31 (CH), 56.21 (2CH_3_), 55.14 (CH_3_), 55.10 (CH_3_). HRMS (ESI) *m*/*z*: calcd for M^+^ C_27_H_26_NO_5_, 444.1805; found, M^+^ 444.1807.

##### (3,4-Bis(3-methoxyphenyl)-1*H*-pyrrol-1-yl)(3,4-dimethoxyphenyl)methanone (**1e**)

White solid 157 mg, yield 35.4%. m.p. 154–156 °C. ^1^H NMR (600 MHz, CDCl_3_) *δ* (ppm) 7.48–7.46 (dd, *J* = 8.3, 1.7 Hz, 1H), 7.44 (s, 2H), 7.41 (d, *J* = 1.6 Hz, 1H), 7.21–7.18 (t, *J* = 7.7 Hz, 2H), 6.97–6.96 (d, *J* = 8.2 Hz, 1H), 6.88–6.87 (d, *J* = 7.7 Hz, 2H), 6.82–6.79 (m, 4H), 3.98 (s, 3H), 3.96 (s, 3H), 3.69 (s, 6H). ^13^C NMR (150 MHz, CDCl_3_) *δ* (ppm) 166.93 (C), 159.50 (2C), 152.94 (C), 149.20 (C), 135.39 (2C), 129.36 (2CH), 128.00 (2C), 124.99 (C), 124.01 (CH), 121.20 (2CH), 120.09(2CH), 114.02 (2CH), 112.82 (2CH), 112.71 (CH), 110.33 (CH), 56.25 (2CH_3_), 55.21 (2CH_3_). HRMS (ESI) *m*/*z*: calcd for M^+^ C_27_H_26_NO_5_, 444.1805; found, M^+^ 444.1803.

##### (3,4-Bis(4-methoxyphenyl)-1*H*-pyrrol-1-yl)(3,4-dimethoxyphenyl)methanone (**1f**)

White solid 145 mg, yield 32.7%. m.p. 156–158 °C. ^1^H NMR (600 MHz, CDCl_3_) *δ* (ppm) 7.47–7.45 (dd, *J* = 8.5, 1.9 Hz, 1H), 7.40–7.39 (d, *J* = 2.2 Hz, 1H), 7.36 (s, 2H), 7.19–7.18 (d, *J* = 8.8 Hz, 4H), 6.97–6.95 (d, *J* = 8.8 Hz, 1H), 6.83–6.82 (d, *J* = 8.8 Hz, 4H), 3.97 (s, 3H), 3.96 (s, 3H), 3.80 (s, 6H). ^13^C NMR (150 MHz, CDCl_3_) *δ* (ppm) 166.91 (C), 158.71 (2C), 152.79 (C), 149.13 (C), 129.75 (4CH), 127.82 (2C), 126.60 (CH), 125.26 (C), 123.91 (2C), 119.39 (2CH), 113.85 (4CH), 112.68 (CH), 110.30 (CH), 56.23 (2CH_3_), 55.32 (2CH_3_). HRMS (ESI) *m*/*z*: calcd for M^+^ C_27_H_26_NO_5_, 444.1805; found, M^+^ 444.1812.

##### (3,4-Bis(2-methoxyphenyl)-1*H*-pyrrol-1-yl)(3,4,5-trimethoxyphenyl)methanone (**1g**)

White solid 187 mg, yield 37.5%. m.p. 180–182 °C. ^1^H NMR (600 MHz, CDCl_3_) *δ* (ppm) 7.52 (s, 2H), 7.22–7.19 (td, *J* = 8.1, 1.6 Hz, 2H), 7.14–7.13 (dd, *J* = 7.6, 1.6 Hz, 2H), 7.09 (s, 2H), 6.87–6.85 (t, *J* = 7.8 Hz, 2H), 6.83–6.82 (d, *J* = 8.2 Hz, 2H), 3.94 (s, 3H), 3.92 (s, 6H), 3.45 (s, 6H). ^13^C NMR (150 MHz, CDCl_3_) *δ* (ppm) 166.17 (C), 155.87 (2C), 152.45 (2C), 140.78 (C), 129.68 (2CH), 127.64 (C), 127.51 (2CH), 125.07 (2C), 123.52 (2C), 119.72 (4CH), 110.16 (2CH), 106.43 (2CH), 55.83 (CH_3_), 54.40 (4CH_3_). HRMS (ESI) *m*/*z*: calcd for M^+^ C_28_H_28_NO_6_, 474.1911; found, M^+^ 474.1911.

##### (3,4-Bis(3-methoxyphenyl)-1*H*-pyrrol-1-yl)(3,4,5-trimethoxyphenyl)methanone (**1h**)

White solid 174 mg, yield 36.7%. m.p. 140–142 °C. ^1^H NMR (500 MHz, CDCl_3_) *δ* (ppm) 7.44 (s, 2H), 7.21–7.19 (m, 2H), 7.07 (s, 2H), 6.87–6.85 (d, *J* = 7.6 Hz, 2H), 6.81–6.80 (m, 4H), 3.95 (s, 3H), 3.93 (s, 6H), 3.69 (s, 6H). ^13^C NMR (125 MHz, CDCl_3_) *δ* (ppm) 166.86 (C), 159.35 (2C), 153.10 (2C), 141.70 (C), 135.07 (2C), 129.26 (2CH), 128.19 (2C), 127.65 (C), 121.02 (2CH), 119.80 (2CH), 113.89 (2CH), 112.72 (2CH), 107.05 (2CH), 61.00 (CH_3_), 56.40 (2CH_3_), 55.06 (2CH_3_). HRMS (ESI) *m*/*z*: calcd for M^+^ C_28_H_28_NO_6_, 474.1911; found, M^+^ 474.1908.

##### (3,4-Bis(4-methoxyphenyl)-1*H*-pyrrol-1-yl)(3,4,5-trimethoxyphenyl)methanone (**1i**)

White solid 178 mg, yield 37.6%. m.p. 163–164 °C. ^1^H NMR (600 MHz, CDCl_3_) *δ* (ppm) 7.36 (s, 2H), 7.19–7.17 (d, *J* = 8.8 Hz, 4H), 7.06 (s, 2H), 6.84–6.83 (d, *J* = 8.3 Hz, 4H), 3.95 (s, 3H), 3.92 (s, 6H), 3.80 (s, 6H). ^13^C NMR (150 MHz, CDCl_3_) *δ* (ppm) 166.24 (C), 158.07 (2C), 152.50 (2C), 141.02 (C), 129.03 (4CH), 127.46 (2C), 127.35 (C), 125.72 (2C), 118.53 (2CH), 113.16 (4CH), 106.47 (2CH), 55.84 (CH_3_), 54.60 (4CH_3_). HRMS (ESI) *m*/*z*: calcd for M^+^ C_28_H_28_NO_6_, 474.1911; found, M^+^ 474.1906.

##### (3,4-Bis(3,4-dimethoxyphenyl)-1*H*-pyrrol-1-yl)(4-methoxyphenyl)methanone (**1j**)

White solid 186 mg, yield 39.3%. ^1^H NMR (500 MHz, CDCl_3_) *δ* (ppm) 7.84–7.83 (d, *J* = 8.6 Hz, 2H), 7.38 (s, 2H), 7.03–7.01 (d, *J* = 8.6 Hz, 2H), 6.86–6.85 (d, *J* = 8.3 Hz, 2H), 6.81–6.78 (m, 4H), 3.90 (s, 3H), 3.87 (s, 6H), 3.69 (s, 6H). ^13^C NMR (125 MHz, CDCl_3_) *δ* (ppm) 166.75 (C), 163.01 (C), 148.44 (2C), 147.97 (2C), 131.92 (2CH), 127.72 (2C), 126.70 (2C), 124.90 (C), 120.82 (2CH), 119.19 (2CH), 113.87 (2CH), 111.99 (2CH), 110.95 (2CH), 55.83 (2CH_3_), 55.67 (2CH_3_), 55.51 (CH_3_). HRMS (ESI) *m*/*z*: calcd for M^+^ C_28_H_28_NO_6_, 474.1911; found, M^+^ 474.1901.

##### (3,4-Bis(3,4-dimethoxyphenyl)-1*H*-pyrrol-1-yl)(3,4-dimethoxyphenyl)methanone (**1k**)

White solid 198 mg, yield 39.4%. ^1^H NMR (500 MHz, CDCl_3_) *δ* (ppm) 7.47–7.45 (d, *J* = 8.2 Hz, 1H), 7.39 (s, 3H), 6.97–6.95 (d, *J* = 8.3 Hz, 1H), 6.85–6.77 (m, 6H), 3.97–3.95 (d, *J* = 7.5 Hz, 6H), 3.87 (s, 6H), 3.69 (s, 6H). ^13^C NMR (125 MHz, CDCl_3_) *δ* (ppm)166.75 (CO), 152.67 (C), 148.96 (C), 148.39 (2C), 147.95 (2C), 127.74 (2C), 126.61 (2C), 124.93 (C), 123.76 (2CH), 120.77 (2CH), 119.22 (CH), 112.47 (CH), 111.92 (2CH), 110.90 (2CH), 110.11 (CH), 56.09 (2CH_3_), 55.80 (2CH_3_), 55.63 (2CH_3_). HRMS (ESI) *m*/*z*: calcd for M^+^ C_29_H_30_NO_7_, 504.2017; found, M^+^ 504.2003.

##### (3,4-Bis(3,4-dimethoxyphenyl)-1*H*-pyrrol-1-yl)(3,4,5-trimethoxyphenyl)methano-ne (**1l**)

White solid 213 mg, yield 39.9%. ^1^H NMR (500 MHz, CDCl_3_) *δ* (ppm) 7.40 (s, 2H), 7.06 (s, 2H), 6.85–6.80 (m, 4H), 6.76 (s, 2H), 3.94 (s, 3H), 3.92 (s, 6H), 3.87 (s, 6H), 3.69 (s, 6H). ^13^C NMR (125 MHz, CDCl_3_) *δ* (ppm) 166.81 (CO), 153.07 (2C), 148.43 (2C), 148.05 (C), 141.61 (2C), 128.10 (C), 127.77 (2C), 126.45 (2C), 120.78 (2CH), 119.11 (2CH), 111.94 (2CH), 110.93 (2CH), 107.02 (2CH), 61.01 (CH_3_), 56.39 (2CH_3_), 55.82 (2CH_3_), 55.65 (2CH_3_). HRMS (ESI) *m*/*z*: calcd for M^+^ C_30_H_32_NO_8_, 534.2122; found, M^+^ 534.2116.

##### (3,4-Bis(3,4,5-trimethoxyphenyl)-1*H*-pyrrol-1-yl)(4-methoxyphenyl)methanone (**1m**)

White solid 209 mg, yield 39.1%. ^1^H NMR (500 MHz, CDCl_3_) *δ* (ppm) 7.84–7.83 (d, *J* = 7.6 Hz, 2H), 7.43 (s, 2H), 7.04–7.02 (d, *J* = 7.6 Hz, 2H), 6.49 (s, 4H), 3.90 (s, 3H), 3.83 (s, 6H), 3.70 (s, 12H). ^13^C NMR (125 MHz, CDCl_3_) *δ* (ppm) 166.75 (CO), 163.14 (C), 152.86 (4C), 136.96 (2C), 131.96 (2CH), 129.38 (2C), 127.81 (2C), 124.63 (C), 119.37 (2CH), 113.94 (2CH), 105.79 (4CH), 60.89 (2CH_3_), 55.95 (4CH_3_), 55.54 (CH_3_). HRMS (ESI) *m*/*z*: calcd for M^+^ C_30_H_32_NO_8_, 534.2122; found, M^+^ 534.2113.

##### (3,4-Bis(3,4,5-trimethoxyphenyl)-1*H*-pyrrol-1-yl)(3,4-dimethoxyphenyl)methanone (**1n**)

White solid 223 mg, yield 39.5%. ^1^H NMR (500 MHz, CDCl_3_) *δ* 7.47–7.45 (dd, *J* = 8.3, 1.9 Hz, 1H), 7.44 (s, 2H), 7.41–7.40 (d, *J* = 1.9 Hz, 1H), 6.98–6.96 (d, *J* = 8.4 Hz, 1H), 6.50 (s, 4H), 3.98 (s, 3H), 3.96 (s, 3H), 3.83 (s, 6H), 3.70 (s, 12H). ^13^C NMR (125 MHz, CDCl_3_) *δ* 166.72 (CO), 152.92 (4C), 152.90(C), 149.13 (C), 137.13 (C), 129.35 (2C), 127.89 (2C), 124.79 (CH), 123.81 (2C), 119.43 (2CH), 112.62 (CH), 110.22 (CH), 105.92 (4CH), 60.89 (2CH_3_), 56.14(CH_3_), 56.13(CH_3_), 55.98 (4CH_3_).

##### (3,4-Bis(3,4,5-trimethoxyphenyl)-1*H*-pyrrol-1-yl)(3,4,5-trimethoxyphenyl)metha-None (**1o**)

White solid 245 mg, yield 42.2%. ^1^H NMR (500 MHz, CDCl_3_) *δ* (ppm) 7.43 (s, 2H), 7.06 (s, 2H), 6.47 (s, 4H), 3.94 (s, 3H), 3.91 (s, 6H), 3.82 (s, 6H), 3.69 (s, 12H). ^13^C NMR (125 MHz, CDCl_3_) *δ* (ppm) 166.82 (CO), 153.14 (2C), 152.91 (4C), 141.79 (C), 137.09 (2C), 129.16 (2C), 128.22 (2C), 127.54 (C), 119.30 (2CH), 107.09 (2CH), 105.81 (4CH), 61.04 (CH_3_), 60.92 (2CH_3_), 56.41 (2CH_3_), 55.95 (4CH_3_). HRMS (ESI) *m*/*z*: calcd for M^+^ C_32_H_36_NO_10_, 594.2334; found, M^+^ 594.2322.

### 3.2. Bioactivity Study

#### 3.2.1. Cell Culture

Rat PC12 cells (adrenal gland; pheochromocytoma) were obtained from Cell Bank of Chinese Academy of Sciences (Shanghai, China). PC12 cells were cultured in Dulbecco’s Modified Eagle’s medium (high glucose, Gibco, San Diego, CA, USA) supplemented with 10% fetal bovine serum (FBS) (Gibco, San Diego, CA, USA), 100 U mL^−1^ penicillin and 100 μg mL^−1^ streptomycin in a 95% humidified atmosphere with 5% CO_2_ at 37 °C. Cells were passaged with trypsin (0.025% EDTA) every 3 days with the subcultivation ratio of 1:6. Cell used was within 5–20 passages.

#### 3.2.2. Cell Viability Assay

The derivatives were first dissolved in DMSO (Solarbio, CN) and then diluted to the test concentration with complete growth media for cellular viability assay.

For cellular viability assay, a 96-wells plate was cultured with 5000 cells well^−1^ in 100 μL complete cell culture medium for 24 h. Then, 100 μL complete medium containing serial concentrations of 2.5, 5, 10, and 20 μM of each compound was added to each well and continued to culture for another 24 h, cell viability was measured by MTT assay as described [28].

For the neuroprotective assay, PC12 cells were cultured into 96-well plates at a density of 5000 cells/well for 24 h. After treating with glutamate of 8 mM for 4 h, the PC12 cells were co-incubated with neolamellarin A derivatives (**1a**–**1o**) at final concentrations of 2.5, 5, 10, and 20 μM for 24 h. Percentage of viable cells were detected. Cell viability was measured by MTT assay.

## 4. Conclusions

Neurotoxicity caused by glutamate widely exists in almost all nervous system diseases, including cerebral hemorrhage (aneurysms, hypertensive encephalorrhagia, etc.), cerebral ischemia, stroke, brain trauma, brain tumors and some neurodegenerative diseases, such as amyotrophic lateral sclerosis, Parkinson’s disease, chronic progressive chorea, etc. In this study, we synthesized and evaluated the protective effects of 15 different 3,4-bisaryl-N-acylated permethylated Neolamellarin A derivatives against glutamate induced PC12 cell apoptosis. The results showed that all of these compounds were effectively against glutamate-induced PC12 cell death. Combined with the previous study results, both the 3,4-bisaryl-N-acylated and 3,4-bisaryl-N-alkylated permethylated Neolamellarin A derivatives showed outstanding neuroprotective activity, and the number and location of methoxy groups had no significant effect on neuroprotective activity. Therefore, we speculate that the skeleton structure of permethylated Neolamellarin A is the key to neuroprotective activity on PC12 cell of these compounds.

## Data Availability

Data are included in the article.

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
