# Peer review of "Protective Effects of Marine Alkaloid Neolamellarin A Derivatives against Glutamate Induced PC12 Cell Apoptosis"

_marinedrugs, 2022, doi:10.3390/md20040262_

Round 1
Reviewer 1 Report
In the study “Protective effects of marine alkaloid Neolamellarin A derivatives against glutamate-induced PC 12 cell apoptosis” Kai Zhang et al. synthesized a series of 15 different 3,4-bisaryl-N-acylated permethylated derivatives of neolamellarin A alkaloid of marine sponge Dendrilla nigra and tested their efficacy in protecting from glutamate induced neurotoxicity in PC-12 rat pheochromocytoma cell culture model in vitro. The authors have previously reported a group of N-alkylated derivatives of Neolamellarin A showed neuroprotective activity at extremely low non-cytotoxic concentrations. The authors also reported that the 3,4-diarylpyrrole-derived alkaloids were very effective in reversing multidrug resistance (MDR) via inhibition of P-glycoprotein-mediated drug efflux. Two synthetic methods were used in the preparation of the novel alkaloids followed by purification and structure confirmation and biological activity testing. Neuroprotective activity of the novel derivatives was evaluated in PC12 cells treated with glutamate. Glutamate is an excitatory neurotransmitter in CNS. Excess glutamate in CNS, however, causes neural apoptotic cell death as a consequence of excess Ca2+ influx activation of proteases such as calpain and parallel generation of ROS, mitochondrial dysfunction. Glutamate-induced neurotoxicity underlies many neurodegenerative diseases. First the authors established non-cytotoxic concentrations of the new derivatives in these cells using MTT cell viability assay in PC 12 cells. They then treated PC12 cells with or without (Control) glutamate in the presence or absence of the newly synthesized derivatives at various concentrations for 24 hrs. The protective activity of the derivatives from glutamate induced apoptotic cell death after 24 hrs. was determined by MTT assay. The authors suggest that these novel agents have neuroprotective activity at non-cytotoxic concentrations and could be developed as therapeutics for neurodegenerative diseases of the CNS.
Comments: Marine organisms are an abundant source of biologically active chemicals with therapeutic properties against a number of pathological conditions including those of CNS. Synthetic chemical studies in uncovering potential compounds with therapeutic activity and structural modification studies to improve biological activity and minimize undesirable side effects should be continued. The study was well thought out and appears to be a continuing effort of the investigators to develop novel neuroprotective compounds from natural products. The use of MTT assay as an apoptotic indicator assay is, however, may not be very reliable unless substantiated by other indicators of cellular apoptosis and their inhibition by test chemicals. MTT assay in this study only indicates cytoprotective activity of Lamellarin A derivatives studies
Reviewer 2 Report
The authors have synthesized new derivatives of Neolamellarin A, which tested only as antagonist of glutamate induced nerve cell apoptosis. The results are very pure, although lot of derivatives were prepared and there is a positive result, more data at the molecular level are needed to support any conclusion. Controls are missing and the manuscript is mainly covered with chemicals details which must be moved in the supplementary session.
Reviewer 3 Report
In this study, PC12 cells were added to glutamate 4 hr before treatment with Neolamellarin compounds. The survival rate of PC12 was about 65 % 4 hr after glutamate treatment. Authors should notice that the survival rate of PC12 cells was over 100 % when some high concentrations (20 umol/L) of 1f, 1h, 1n, and 1o were treated (Figure 3). However, treated PC12 cells alone with these compounds did not affect the survival rate (Figure 2). Authors should explain this phenomenon.
1a, 1b, 1c will induce cell proliferation? However, only a high concentration of 1C will inhibit glutamate cause cell death?
Round 2
Reviewer 2 Report
The presentation has been improved, but the scientific impact still remains low. More experiments are need to investigate at the molecular level the reason of inhibition glutamate induced apoptosis